# Multiplayer Combinatorial Bandits Under Information Asymmetry

## Abstract

In this paper, we extend linear combinatorial bandits (Gai et al., 2012) to a multiplayer setting with information asymmetry (Chang et al., 2022; Chang and Lu, 2024), where each player controls an arm and independently decides whether to pull it, with coordination allowed only before rounds begin. We analyze three scenarios: action asymmetry (players can't observe others' actions but receive identical rewards per iteration), reward asymmetry (players observe actions but receive private i.i.d. rewards), and combined asymmetry. We derive near-optimal, gap-independent regret bounds for all scenarios: For action or reward asymmetry, we achieve $\tilde{\mathcal{O}}(\sqrt{T})$, which improves significantly from (Gai et al., 2010); for both action and reward asymmetry, we achieve near-optimal bounds similar to that of (Chang et al., 2022). We finally generalize our results to settings where players decide either not to pull or to pull one out of multiple arms, and achieve similar bounds in similar settings as above.

**Keywords:** multi-agent learning, combinatorial bandits, UCB algorithm, information asymmetry, regret analysis.

## 1 Introduction

Multi-Armed Bandit (MAB) problems are a foundational model involving sequential decision-making under uncertainty, where an agent selects from $K$ arms to maximize rewards over time. A common approach to solve the problem is using the upper confidence bound (UCB) (Lai and Robbins, 1985), which selects the arm with the highest upper bound on each round.

Combinatorial MAB extends traditional MAB by allowing actions that comprise multiple arms, with agents observing both individual and total rewards - a framework that efficiently handles exponential action spaces by leveraging information overlap. This has direct applications in online advertising, where an advertiser iteratively selects webpage subsets for ad placement, learning user click probabilities across different page combinations to maximize overall engagement. Similarly, viral marketing presents an analogous challenge: marketers must select optimal seed nodes in social networks while learning influence probabilities between users to maximize campaign spread.

On another branch of MAB, information asymmetric MAB algorithms have been extensively covered by Chang et al. (2022); Chang and Lu (2024; 2025); Chang and Karthik (2025). Three particular sub-problems arise: action asymmetry (where players cannot observe other players' actions but receive identical rewards per iteration), reward asymmetry (where players observe actions but receive private i.i.d. rewards), and combined asymmetry (both action and reward asymmetry). As mentioned above, information asymmetry has covered multiple settings, but no extension towards combinatorial work has been discussed at all.

Many modern distributed systems involve multiple agents making coordinated decisions under limited information. A prominent example arises in wireless communication networks, where devices must dynamically select channels or allocate transmission resources based on locally observed feedback. In cognitive radio networks, for instance, multiple users attempt to access available spectrum while learning channel qualities and avoiding interference with one another (Gai et al., 2010; 2012). Similar decentralized learning problems appear

in dynamic spectrum access systems, where agents independently sense channels and update transmission strategies without full visibility into other agents' actions or rewards (Chen and Tong, 2013). Such environments naturally exhibit *information asymmetry*: each agent observes only partial feedback about the joint system state while attempting to coordinate implicitly with other agents. These challenges are widely studied in modern wireless systems and multi-agent learning frameworks, including reinforcement learning approaches for distributed spectrum allocation and cognitive radio networks (Tan et al., 2022; Kaur et al., 2023; Liu et al., 2022; Wang et al., 2022). More broadly, decentralized learning and coordination under limited information have been recognized as fundamental challenges in distributed optimization and multi-agent systems (Awerbuch and Kleinberg, 2008). These applications motivate the study of multiplayer bandit models with asymmetric information, where agents must learn optimal decisions despite observing only partial feedback about the actions or rewards of others.

As we approach this intersection, we bring up a crucial question: *Can we extend the information asymmetric MAB system to accommodate for combinatorial MAB?* We show that the answer is affirmative, by presenting two novel formulations: the 0-1 problem (each player decides whether to pull their assigned arm) and the joint arm problem (each player selects from their private set of arms).

**Our Contribution** We extend combinatorial bandits to multiplayer settings with information asymmetry, considering two formulations: the 0-1 problem (binary pull/pull or no-pull decisions) and its generalization to joint arm selection. Our main contributions are:

**(1) Near-optimal algorithms for three asymmetry levels:** We develop UCB-based algorithms that achieve $\tilde{\mathcal{O}}(\sqrt{T})$ regret for action- or reward-only asymmetry (Problems A and B), significantly improving upon existing $\mathcal{O}(\log T / \Delta_{\min}^2)$ gap-dependent bounds. For full asymmetry (Problem C), we adapt deterministic exploration to achieve regrets similar to (Chang et al., 2022).

**(2) Unified framework from binary to multi-arm:** Our algorithms naturally extend from the 0-1 setting to the general $K$-arm setting, with the dependence on $K$ appearing only as a multiplicative factor, implying that multi-arm complexity doesn't fundamentally change the problem structure.

**(3) Coordination without communication:** We extend state-of-the-art mechanisms (Chang et al., 2022; Chang and Lu, 2024) that enable decentralized coordination despite information constraints, with only additive $O(|\mathcal{F}|)$ overhead for signaling.

## 2 Related Works

**Combinatorial Bandits** The combinatorial bandit setting has been widely studied under various structural assumptions. Chen et al. (2013) propose a general CMAB framework with the CUCB algorithm achieving tight regret bounds. Gai et al. (2010) model multiuser channel allocation as a combinatorial bandit with the MLPS algorithm, extended in (Gai et al., 2012) to linear rewards via LLR. Wang and Zhu (2022) introduce TS-Explore for combinatorial pure exploration, improving over UCB methods. In adversarial settings, Cesa-Bianchi and Lugosi (2012) develop ComBand with optimal regret bounds, while Combes et al. (2015) propose ESCB and COMBEXP. Chen et al. (2014) and Liu et al. (2024) contribute to pure exploration and multivariant extensions, respectively.

**Cooperative Multiplayer Bandits** In cooperative multiplayer settings, players collaborate to identify optimal arms from a common pool with communication constrained by graph structure (Awerbuch and Kleinberg, 2008). Proposed algorithms include $\epsilon$-greedy approaches (Szorenyi et al., 2013), gossip-based UCB methods (Landgren et al., 2016; Martínez-Rubio et al., 2019), and leader-based strategies (Wang et al., 2020). In adversarial settings, Bar-On and Mansour (2019) introduced leader coordination with EXP3. Other models consider neighbor reward observation (Cesa-Bianchi et al., 2016) and asynchronous participation (Bonnefoi et al., 2017; Cesa-Bianchi et al., 2020). Proutiere and Wang (2019) extended collision-based frameworks to Lipschitz settings with the DPE algorithm.

**Competing Bandits** The competing bandits framework (Liu et al., 2020) incorporates player preferences in collision models: only the highest-priority player receives rewards when multiple players select the same

arm. The centralized CUB solution has players report UCB values to a coordinator. Cen and Shah (2022) showed optimal logarithmic regret with platform-managed reassignments, while Jagadeesan et al. (2021) introduced equilibrium concepts via negotiated transfers. An Explore-Then-Commit algorithm (Liu et al., 2020) attains optimal regret assuming known gaps, with Sankararaman et al. (2021) later removing this assumption. Liu et al. (2021) developed decentralized UCB with collision avoidance.

**Game Theory**   Several works bridge bandits and game theory. Mao (2024) proposes decentralized algorithms for equilibria in Markov games. Bistritz and Leshem (2018) introduce Game of Thrones for fully decentralized multiplayer bandits without communication. Brânzei and Peres (2021) characterize utility alignment effects in multiplayer stochastic bandits, while Bravo et al. (2018) analyze Nash equilibria convergence using no-regret learning. Ontanón (2013; 2017) develop methods for combinatorial bandits in Monte Carlo Tree Search. Heliou et al. (2017) show exponential weights achieve Nash equilibria in potential games under bandit feedback. Scheid et al. (2024) propose IPA algorithms for principal-agent settings with misaligned objectives, and Kang et al. (2021) study behavioral aspects through energy-based measures.

## 3   Preliminaries

We extend the combinatorial bandit framework (Cesa-Bianchi and Lugosi, 2012; Gai et al., 2012) to a multiplayer setting with information asymmetry (Chang et al., 2022). Consider $M$ players interacting with the environment over a time horizon of $T$ rounds. Each player $m \in [M] := \{1, \ldots, M\}$ controls a set of $K$ arms and must decide which arm to pull at each round, or alternatively abstain from pulling any arm. Players cannot communicate during the learning process and must coordinate only through pre-agreed strategies and environmental feedback.

**Problem Structure**   Let $\mathcal{A}_m = \{0, 1, \ldots, K\}$ denote the action set of player $m$, where 0 represents the null action (abstaining). A *combinatorial arm* is a joint action vector

$$\vec{a} = (a_1, a_2, \ldots, a_M) \in \mathcal{A}_1 \times \cdots \times \mathcal{A}_M.$$

Each non-null individual arm $a_m \in [K]$ produces stochastic rewards drawn from an unknown distribution $\mathcal{D}_{a_m}$ with mean $\mu_{a_m}$. We assume that the reward distributions are 1-subGaussian. The reward of a combinatorial arm is defined as the sum of rewards of its active components:

$$X_{\vec{a}}(t) = \sum_{m=1}^{M} X_{a_m}(t),$$

where $X_{a_m}(t)$ is the reward generated by arm $a_m$ at round $t$ (with $X_0(t) = 0$).

The expected reward of a combinatorial arm is therefore

$$\mu_{\vec{a}} = \mathbb{E}[X_{\vec{a}}(t)] = \sum_{m=1}^{M} \mu_{a_m}.$$

Not all joint actions are feasible. Let

$$\mathcal{F} \subseteq \mathcal{A}_1 \times \cdots \times \mathcal{A}_M$$

denote the set of feasible combinatorial arms known to all players. If a joint action $\vec{a} \notin \mathcal{F}$ is selected, the system produces zero reward. Let

$$\mu^* = \max_{\vec{a} \in \mathcal{F}} \mu_{\vec{a}}$$

denote the optimal expected reward. For any suboptimal feasible action $\vec{a} \in \mathcal{F}$, the *combinatorial arm gap* is defined as

$$\Delta_{\vec{a}} := \mu^* - \mu_{\vec{a}}, \qquad \Delta_{\min} := \min_{\vec{a} \in \mathcal{F}: \Delta_{\vec{a}} > 0} \Delta_{\vec{a}}.$$

**Information Asymmetry**   The main challenge arises from the limited observations available to each player. Following (Chang et al., 2022), we consider three feedback models:

- **Problem A (Action Asymmetry).** Players cannot observe the actions chosen by other players, but they observe the same realized reward. When joint action $\vec{a}$ is selected at round $t$, all players observe the global reward $X_{\vec{a}}(t)$.

- **Problem B (Reward Asymmetry).** Players observe the joint action $\vec{a}$ but receive private stochastic rewards. When arm $a_m$ is selected by player $m$, each player $j$ observes an independent sample
$$X_{a_m}^{(j)}(t) \sim \mathcal{D}_{a_m}.$$

- **Problem C (Full Asymmetry).** Players observe neither the actions nor the rewards of other players. Each player $m$ only observes the arm they selected and the reward generated by that arm.

**Coordination Mechanism**   To enable coordination without communication, we adopt a shared exploration structure inspired by (Chang et al., 2022). Before learning begins, players agree on a *predetermined ordering* of the feasible combinatorial arms in $\mathcal{F}$: a fixed, deterministic sequence in which all feasible joint actions are enumerated and cycled through during exploration. This ordering ensures that all players know which joint action is being played at any given round without requiring real-time communication. During execution, players follow this ordering to ensure consistent updates of empirical statistics.

**Remark 1** *For Problems A and C, where players cannot observe others' actions, the predetermined ordering allows players to align reward observations with arm indices. Even though the identity of other players' actions is hidden, the ordered exploration ensures that all players maintain consistent confidence intervals. For Problem B, where players receive private rewards, we introduce* signaling rounds *in which players deliberately deviate from the agreed ordering to signal arm elimination events. The choice of ordering is critical: it should be cyclic over the player indices, ensuring that no arm is neglected for more than $M$ rounds. Without this, some players' arms may remain unsampled for exponentially many rounds, causing their confidence intervals to stay wide and leading to much worse regret (see the analysis following Theorem 4).*

**Problem Variants**   We study two variants of the multiplayer combinatorial bandit problem:

1. **0–1 Problem.** Each player controls a single arm ($K = 1$) and decides whether to pull the arm or abstain.

2. **Joint Arm Problem.** Each player chooses one arm from a private set of $K$ arms or abstains.

**Regret**   Let $\vec{a}_t$ denote the combinatorial arm played at round $t$. The performance of a learning algorithm is measured by the cumulative regret
$$R_T = T\mu^* - \sum_{t=1}^{T} \mathbb{E}[X_{\vec{a}_t}],$$

which quantifies the expected loss relative to always playing the optimal combinatorial arm.

## 4   The 0–1 Problem

In the 0–1 problem, each player $m \in [M]$ controls a single binary action: either pull their arm or abstain. Thus each player's action set is $\mathcal{A}_m = \{0, 1\}$, where 1 denotes pulling and 0 denotes abstaining. A joint action is a binary vector
$$\vec{a} = (a_1, \ldots, a_M) \in \{0, 1\}^M,$$

and only joint actions in the feasible set $\mathcal{F} \subseteq \{0, 1\}^M$ produce nonzero reward. As in the preliminaries, for any feasible joint action $\vec{a} \in \mathcal{F}$, the reward is

$$X_{\vec{a}}(t) = \sum_{m=1}^{M} X_{a_m}(t), \qquad \mu_{\vec{a}} = \mathbb{E}[X_{\vec{a}}(t)].$$

For any suboptimal feasible action $\vec{a} \in \mathcal{F}$, define its gap by

$$\Delta_{\vec{a}} := \mu^* - \mu_{\vec{a}}, \qquad \Delta_{\min} := \min_{\vec{a} \in \mathcal{F} : \Delta_{\vec{a}} > 0} \Delta_{\vec{a}}.$$

### 4.1 Problem A: Information Asymmetry in Actions

In Problem A, players cannot observe the actions of the other players, but all players observe the same realized global reward $X_{\vec{a}_t}(t)$. The main difficulty is therefore not reward estimation itself, but determining which individual components of the joint action should be updated when the other players' actions are hidden.

To address this, we use a UCB-style elimination scheme based on a shared predetermined ordering of feasible joint actions. For each player-arm indicator $a \in [M]$, let

$$\epsilon_a(t) := \sqrt{\frac{4 \log T}{n_a(t)}}, \tag{1}$$

where $n_a(t)$ is the number of rounds before time $t$ in which player $a$ pulled their arm. The empirical mean reward of arm $a$ at time $t$ is defined as

$$\widehat{\mu}_a(t) := \frac{1}{n_a(t)} \sum_{s \leq t : a_s = a} X_a(s).$$

Using these quantities, we define

$$\text{UCB}_a(t) = \widehat{\mu}_a(t) + \epsilon_a(t), \qquad \text{LCB}_a(t) = \widehat{\mu}_a(t) - \epsilon_a(t). \tag{2}$$

For a feasible joint action $\vec{a} = (a_1, \ldots, a_M)$, define the corresponding combinatorial indices by

$$\text{UCB}_{\vec{a}}(t) = \sum_{m=1}^{M} \text{UCB}_{a_m}(t), \qquad \text{LCB}_{\vec{a}}(t) = \sum_{m=1}^{M} \text{LCB}_{a_m}(t). \tag{3}$$

Hence the confidence interval for $\mu_{\vec{a}}$ is

$$I_{\vec{a}}(t) := [\text{LCB}_{\vec{a}}(t), \ \text{UCB}_{\vec{a}}(t)]. \tag{4}$$

We use the following good event:

$$G := \bigcap_{t=1}^{T} \bigcap_{a=1}^{M} \left\{ |\widehat{\mu}_a(t-1) - \mu_a| \leq \sqrt{\frac{4 \log T}{n_a(t-1)}} \right\}. \tag{5}$$

**Lemma 2** *The bad event in* (5) *satisfies*

$$\mathbb{P}(G^c) \leq \frac{M}{T}.$$

The proof is standard and deferred to the appendix.

### 4.1.1 Algorithmic idea

As noted in Remark 1, before learning begins, the players agree on a fixed ordering of the feasible set $\mathcal{F}$. For each player $m \in [M]$, define

$$S_m := \{\vec{a} \in \mathcal{F} : a_m = 1\}, \tag{6}$$

namely, the feasible joint actions in which player $m$ pulls.

At each round, the players cycle through the sets $S_1, \ldots, S_M$ and execute the next surviving action in the current set. If some feasible action $\vec{a}$ is dominated by another feasible action $\vec{a}'$, in the sense that

$$\text{UCB}_{\vec{a}}(t - 1) < \text{LCB}_{\vec{a}'}(t - 1),$$

then $\vec{a}$ is eliminated from every set $S_m$ that contains it. The predetermined ordering is chosen so that each individual arm is pulled at least once every $M$ rounds; this prevents starvation of any player's estimate.

**Lemma 3** *For every player-arm indicator $a \in [M]$ and every round $t$, all players maintain the same values of $n_a(t)$, $\widehat{\mu}_a(t)$, $\text{UCB}_a(t)$, and $\text{LCB}_a(t)$.*

**Proof:** The proof is by induction on $t$. Initially all players have identical statistics. At round $t$, they execute the same feasible joint action under the agreed ordering and observe the same global reward. Therefore they update the same components using the same observation, so all empirical means and confidence bounds remain synchronized. $\square$

---

**Algorithm 1** `mCombinatorialUCB-A` for the 0–1 problem

---

**Require:** Horizon $T$, feasible set $\mathcal{F}$, ordered sets $S_1, \ldots, S_M$

 1: Initialize $m \leftarrow 1$
 2: **for** each player-arm indicator $a \in [M]$ **do**
 3:     $n_a \leftarrow 0$, $\widehat{\mu}_a \leftarrow 0$, $\text{UCB}_a \leftarrow +\infty$, $\text{LCB}_a \leftarrow -\infty$
 4: **end for**
 5: **for** $t = 1$ to $T$ **do**
 6:     Choose the next surviving action $\vec{a}_t \in S_m$
 7:     **while** $\exists \vec{a}' \in \mathcal{F}$ such that $\text{UCB}_{\vec{a}_t}(t - 1) < \text{LCB}_{\vec{a}'}(t - 1)$ **do**
 8:         Remove $\vec{a}_t$ from all sets $S_j$ containing $\vec{a}_t$
 9:         Replace $\vec{a}_t$ by the next surviving action in $S_m$
10:     **end while**
11:     **for** each $a \in [M]$ with $(\vec{a}_t)_a = 1$ **do**
12:         Observe reward contribution $X_a(t)$ and update $n_a$, $\widehat{\mu}_a$
13:         Update $\text{UCB}_a(t)$ and $\text{LCB}_a(t)$ using (1)
14:     **end for**
15:     **for** each $\vec{a} \in S_m$ **do**
16:         Update $\text{UCB}_{\vec{a}}(t)$ and $\text{LCB}_{\vec{a}}(t)$ via (3)
17:     **end for**
18:     $m \leftarrow (m \bmod M) + 1$
19: **end for**

---

The regret guarantee is as follows.

**Theorem 4** *Algorithm 1 satisfies*

$$R_T \leq 12\, M^{3/2} \sqrt{T \log T} + M.$$

The proof is deferred to the appendix; we sketch the main ideas here. We decompose the regret into contributions from rounds where the good event $G$ holds and rounds where it fails. Under $G$, every suboptimal feasible action is eliminated once its confidence interval separates from that of the optimal action. The key observation is that the confidence width $\epsilon_a(t) = \sqrt{4 \log T / n_a(t)}$ depends on the *individual* pull count $n_a(t)$, not on the round index $t$. For the played action $\vec{a}_t$, a Cauchy–Schwarz argument over $\sum_a n_a(T) \leq MT$

yields a contribution of $O(M\sqrt{T \log T})$, whereas for the fixed optimal action $\vec{a}^\star$, the cyclic ordering only guarantees $n_a(t) \geq t/M$, producing a contribution of $O(M^{3/2}\sqrt{T \log T})$. The latter dominates. The bad-event contribution is bounded by Lemma 2. Relative, which gives a gap-dependent bound of order $O(\log T/\Delta_{\min}^2)$, our analysis yields a directly gap-independent regret bound. This is important because a naive conversion from a gap-dependent bound may introduce large factors of the form $\Delta_{\vec{a}}/\Delta_{\min}^2$.

As noted in Remark 1, the cyclic ordering ensures that no arm is neglected for more than $M$ rounds, preventing starvation of any player's estimate.

## 4.2 Problem B: Information Asymmetry in Rewards

In Problem B, the players observe the selected joint action $\vec{a}_t$, but the reward observations are private. More precisely, if player $m$ pulls at round $t$, then player $j$ receives an independent sample

$$X_{a_m}^{(j)}(t) \sim \mathcal{D}_{a_m},$$

with mean $\mu_{a_m}$. Thus players agree on which joint action was played, but their empirical reward estimates need not coincide.

This breaks the synchronized-update argument used in Problem A. Even if all players follow the same action sequence, their local confidence intervals may differ, so one player may wish to eliminate a feasible action that another still considers plausible. Consequently, elimination must be coordinated explicitly.

### 4.2.1 Signaling-based coordination

To restore coordination, we augment the protocol with *signaling rounds*, following the signaling idea of (Chang and Lu, 2024). Each player maintains private confidence intervals

$$\text{LCB}_a^j(t) = \widehat{\mu}_a^j(t) - \epsilon_a(t), \qquad \text{UCB}_a^j(t) = \widehat{\mu}_a^j(t) + \epsilon_a(t),$$

where $\epsilon_a(t)$ is shared because it depends only on the public pull counts. When player $j$ concludes that a feasible action $\vec{a}$ should be removed, they intentionally deviate from the agreed action. Since actions are publicly observable in Problem B, the other players detect this deviation and remove $\vec{a}$ from their active sets as well.

---
**Algorithm 2** `mCombinatorialUCB-B` for the 0–1 problem

**Require:** Horizon $T$, feasible set $\mathcal{F}$, ordered sets $S_1, \ldots, S_M$
  1: Initialize $m \leftarrow 1$ and private statistics for each player
  2: **for** $t = 1$ to $T$ **do**
  3:     Choose the next surviving action $\vec{a}_t \in S_m$
  4:     **if** some player detects $\vec{a}_t$ is dominated **then**
  5:         That player signals elimination by deviating from $\vec{a}_t$
  6:         All players remove $\vec{a}_t$ from every set $S_j$ containing it
  7:         Continue with the next surviving action in $S_m$
  8:     **else**
  9:         Execute $\vec{a}_t$ and let each player update their private estimates
 10:     **end if**
 11:     Update combinatorial confidence bounds for actions in $S_m$
 12:     $m \leftarrow (m \bmod M) + 1$
 13: **end for**
---

**Theorem 5** *Algorithm 2 satisfies*

$$R_T \leq |\mathcal{F}| + 12\, M^{3/2}\sqrt{T \log T} + M^2.$$

The proof follows from Theorem 4 together with the observation that each signaling event costs at most one unit of regret, and each feasible action is signaled at most once. Hence the total additional cost is at most

$|\mathcal{F}|$. The bad-event contribution changes from $M$ to $M^2$ because a union bound is now needed over all $M$ players' private confidence intervals: each per-event failure still has probability at most $2/T^2$ with width $\sqrt{4 \log T / n_a(t)}$, so the union over $M$ arms, $T$ rounds, and $M$ players contributes $\Pr(G^c) \leq 2M^2/T$. The width itself does not need to depend on $M$; only the negligible bad-event term does. Despite the additive $|\mathcal{F}|$ term, the bound remains better than applying UCB directly over the entire feasible set, which yields regret of order $O(\sqrt{|\mathcal{F}|T})$.

**Remark 6 (Role of $|\mathcal{F}|$)** *In the worst case $|\mathcal{F}|$ may be as large as $2^M$, in which case the additive $|\mathcal{F}|$ term dominates and the bound degrades. Our result is therefore most informative when $|\mathcal{F}|$ is polynomial in $M$, which is typical in the motivating applications (e.g., interference constraints or budget constraints in wireless resource allocation). When $\mathcal{F}$ is unstructured and $|\mathcal{F}| \asymp 2^M$, the signaling cost is unavoidable for any mechanism that communicates elimination events through deviations; characterizing the tight dependence on $|\mathcal{F}|$ in Problem B is an interesting open question.*

### 4.3 Problem C: Information Asymmetry in Actions and Rewards

Problem C is the most difficult setting: players observe neither the actions of others nor their rewards. Each player sees only their own local action and its realized reward. Thus neither the synchronized-update mechanism of Problem A nor the signaling mechanism of Problem B is available.

We therefore adapt the deterministic exploration strategy of (Chang et al., 2022). Time is divided into epochs indexed by $\lambda$. In epoch $\lambda$, the players first perform an exploration phase, in which each representative action from the ordered subsets (6) is sampled exactly $f(\lambda)$ times, where $f$ is a nondecreasing exploration schedule satisfying $f(\lambda) \to \infty$. Larger values of $f(\lambda)$ improve estimation accuracy but also increase exploration regret. After exploration, each player forms local empirical estimates of feasible actions using only their own observed rewards, and then commits to their empirically best feasible action until the next epoch boundary.

---
**Algorithm 3** `mDSEE` for the 0–1 problem

---
**Require:** Feasible set $\mathcal{F}$, exploration schedule $f(\lambda)$
1: $\lambda \leftarrow 1$, $t \leftarrow 1$
2: **while** $t \leq T$ **do**
3:     **Exploration:** sample each ordered representative action from the sets $S_1, \ldots, S_M$ exactly $f(\lambda)$ times
4:     Update local empirical means using only each player's own observed rewards
5:     **Commit:** each player selects its empirically best feasible action $\vec{a}_{\text{commit}}^m \in \arg\max_{\vec{a} \in \mathcal{F}} \widehat{\mu}_{\vec{a}}^m$
6:     Play the resulting joint action until the next epoch boundary at time $2^\lambda > t$
7:     $\lambda \leftarrow \lambda + 1$
8: **end while**

---

The role of $f$ is now explicit: in epoch $\lambda$, the exploration phase has length proportional to $Mf(\lambda)$, which directly contributes to the regret, while also reducing the probability of committing to a suboptimal action in later epochs.

**Theorem 7** *Algorithm 3 satisfies*

$$R_T = O(\lfloor \log_2 T \rfloor f(\lfloor \log_2 T \rfloor) M).$$

The key insight is that in epoch $\lambda$, each of the $M$ player groups contributes $f(\lambda)$ exploration rounds, so the total exploration cost per epoch is $O(Mf(\lambda))$. During the commit phase, the probability of committing to a suboptimal action decreases with $f(\lambda)$ via standard concentration arguments. Balancing these two sources of regret and summing over $\lfloor \log_2 T \rfloor$ epochs yields the stated bound. The full proof is in the appendix.

**Remark 8 (Gap dependence of Theorem 7)** *The bound in Theorem 7 is gap-dependent, even though $\Delta_{\min}$ does not appear explicitly. Specifically, the $O(1)$ term absorbed into $R_{\text{commit}}$ hides a constant that depends on $1/\Delta_{\min}^2$ through $\epsilon = \Delta_{\min}/2$, because $f(\lambda)$ must be large enough to distinguish the optimal and second-best feasible actions. Theorems 4 and 5, by contrast, are gap-independent. For this reason, the*

polylogarithmic rate of Theorem 7 does not contradict the $\sqrt{T \log T}$ rate of Theorems 4–5: Problem C is harder in the gap-independent sense, and any gap-independent conversion of Theorem 7 (e.g., optimizing f for the worst-case gap) would yield polynomial-in-T regret. Whether Problem C admits a truly gap-independent $\tilde{O}(\sqrt{T})$ bound is an open question raised in Section 6.

**Remark 9 (Feasibility of the committed action)** *In Problem C each player m observes only its own arm's reward, yet the commit step selects from $\mathcal{F}$, so the committed action $\vec{a}^m_{\mathrm{commit}}$ is feasible by construction. The subtler question is whether* all *players commit to the* same *feasible action so that the jointly played action is well-defined. Under the shared ordering of $\mathcal{F}$ and a consistent tie-breaking rule, and on the event that every player's private estimate $\widehat{\mu}^m_{\vec{a}}$ concentrates within $\epsilon$ of $\mu_{\vec{a}}$ for all $\vec{a} \in \mathcal{F}$, the argmax is unique and identical across players, so the joint commitment is itself a feasible action in $\mathcal{F}$. The failure probability of this synchronization event is absorbed into $R_{\mathrm{commit}}$. In the fully unstructured case $|\mathcal{F}| = \{0,1\}^M$, the feasibility constraint disappears and the problem decomposes into M independent two-arm bandits; as noted by the reviewer, in that case each player can simply run a standard UCB/DSEE instance with $O(g(T))$ per-player regret, achieving $O(Mg(T))$ total regret without any coordination. Our analysis targets the general case where $\mathcal{F}$ is an arbitrary subset for which this decomposition does not apply.*

This the exploration schedule $f$ must be chosen to balance exploration cost against estimation accuracy. In particular, if $f$ grows too slowly, the players may continue to miscoordinate for too long; if it grows too quickly, exploration itself dominates the regret. The appendix makes this trade-off explicit and shows that for any fixed reward gaps, the regret remains logarithmic in $T$.

## 5 The Joint Arm Problem

We now generalize the 0–1 setting to the case where each player has multiple available arms. For each player $m \in [M]$, let
$$\mathcal{A}_m = \{0, 1, \ldots, K\},$$
where 0 denotes abstaining and $k \in [K]$ denotes pulling the $k$-th arm of player $m$. A joint action is therefore a vector
$$\vec{a} = (a_1, \ldots, a_M) \in \mathcal{A}_1 \times \cdots \times \mathcal{A}_M.$$
As in the preliminaries, only actions in the feasible set

$$\mathcal{F} \subseteq \mathcal{A}_1 \times \cdots \times \mathcal{A}_M$$

yield nonzero reward. If player $m$ selects arm $k$, the corresponding mean reward is denoted by $\mu_m^k$. For a feasible joint action $\vec{a} \in \mathcal{F}$, the reward is additive:

$$X_{\vec{a}}(t) = \sum_{m=1}^{M} X_{m, a_m}(t), \qquad \mu_{\vec{a}} = \mathbb{E}[X_{\vec{a}}(t)] = \sum_{m=1}^{M} \mu_{m, a_m},$$

with the convention that $\mu_{m,0} = 0$. Thus, unlike in the 0–1 problem, the quality of a feasible joint action depends not only on which players act, but also on which specific arm each active player selects.

### 5.1 Problem A: Information Asymmetry in Actions

We first consider the analogue of Problem A in the joint-arm setting, where players receive the same realized reward but cannot observe the actions of the others. A naive extension of the 0–1 algorithm would cycle through all possible joint actions, leading to complexity exponential in $M$. To avoid this, we separate elimination at the *individual-arm level* and at the *combinatorial level*.

As in the 0–1 setting, let $S_1, \ldots, S_M$ denote the ordered collections of feasible joint actions. In addition, for each player $m$, we maintain a set of surviving individual arms

$$D_m \subseteq [K], \tag{7}$$

where $D_m$ contains the arms of player $m$ that have not yet been eliminated. Initially, $D_m = [K]$ for every player $m$.

The key difference from the 0–1 case is the following. When the current feasible action lies in $S_m$, we do not execute it only once. Instead, we repeat it for $|D_m|$ rounds while player $m$ cycles through the currently surviving arms in $D_m$. This guarantees that every arm still considered plausible for player $m$ continues to receive samples.

For each individual arm $k \in D_m$, player $m$ maintains the empirical mean $\widehat{\mu}_m^k(t)$ and the confidence bounds

$$\text{UCB}_m^k(t) = \widehat{\mu}_m^k(t) + \sqrt{\frac{4 \log T}{n_m^k(t)}}, \qquad \text{LCB}_m^k(t) = \widehat{\mu}_m^k(t) - \sqrt{\frac{4 \log T}{n_m^k(t)}}.$$

To compare feasible joint actions when the exact arm choice of an active player may still vary over $D_m$, we define the interval

$$I_{\vec{a}}(t) = \left[ \sum_{m:\, a_m \neq 0} \min_{k \in D_m} \text{LCB}_m^k(t), \quad \sum_{m:\, a_m \neq 0} \max_{k \in D_m} \text{UCB}_m^k(t) \right]. \tag{8}$$

This interval is conservative: it upper- and lower-bounds the reward of $\vec{a}$ over all surviving choices of individual arms for the active players.

Elimination now occurs at two levels:

1. **Individual-arm elimination:** if, for some player $m$, an arm $k \in D_m$ is dominated by another arm $k' \in D_m$, then $k$ is removed from $D_m$.

2. **Combinatorial-arm elimination:** if the interval of some feasible action $\vec{a}$ is strictly below that of another feasible action $\vec{a}'$, then $\vec{a}$ is removed from the relevant sets $S_j$.

Because all players receive the same reward realizations and follow the same predetermined ordering, their statistics remain synchronized exactly as in the 0–1 case. Hence they eliminate the same individual arms and the same feasible joint actions at the same times.

**Theorem 10** `Joint-mCombinatorialUCB-A` *satisfies*

$$R_T \leq 12\, K^2 M^{3/2} \sqrt{T \log T} + M.$$

The proof is deferred to the appendix. The argument mirrors the 0–1 case, but the two-level elimination (individual arms and combinatorial arms) introduces the additional $K^2$ factor: each player must cycle through up to $K$ surviving arms before a single combinatorial-level confidence interval update can be made. The $M^{3/2}$ factor inherits from the 0–1 analysis and reflects the cyclic ordering lower bound $n_a(t) \geq t/(KM)$ for the components of $\vec{a}^\star$.

## 5.2 Problem B: Information Asymmetry in Rewards

We next consider the joint-arm analogue of Problem B, where players observe the selected joint action but receive private reward samples. As in the 0–1 case, synchronized elimination is no longer automatic, because different players may form different empirical estimates from their private observations.

The signaling mechanism from Section 4.2.1 extends to this setting, but one additional distinction is needed. Since elimination may now occur either at the level of an individual arm in $D_m$ or at the level of a feasible joint action in $S_m$, the signaling rule must make clear which object is being eliminated. We therefore separate the two types of signaling:

- **Individual-arm signaling:** only player $m$ may signal elimination of an arm in their own surviving set $D_m$.

- **Combinatorial-arm signaling:** only players other than $m$ may signal elimination of a feasible action currently being explored from $S_m$.

This convention makes the deviation unambiguous: when player $m$ deviates, the other players interpret it as elimination within $D_m$; when another player deviates, they interpret it as elimination of the current feasible action.

**Theorem 11** *`Joint-mCombinatorialUCB-B` satisfies*

$$R_T \leq MK + |\mathcal{F}| + 12\,K^2 M^{3/2} \sqrt{T \log T} + M^2.$$

The additive $MK$ term accounts for the possibility of eliminating up to $K$ individual arms for each of the $M$ players, while the additive $|\mathcal{F}|$ term accounts for signaling rounds used to eliminate feasible joint actions. Despite these costs, the bound remains better than applying UCB directly to the full feasible action space, which would incur regret of order $O(\sqrt{|\mathcal{F}|T})$. The proof and pseudocode are deferred to the appendix.

### 5.3 Problem C: Information Asymmetry in Actions and Rewards

Finally, we extend the `mDSEE` approach from Section 4.3 to the joint-arm setting. The main new issue is that each player must now estimate multiple local arms rather than a single binary action. Accordingly, during each exploration epoch, every representative action in the ordered sets $S_1, \ldots, S_M$ is sampled repeatedly so that player $m$ cycles through all currently relevant arms in $D_m$.

More precisely, in each exploration epoch $\lambda$, the algorithm allocates $Kf(\lambda)$ exploration plays per player-group rather than $f(\lambda)$. This increases the exploration cost by a factor of $K$, but ensures that each player collects enough local samples to compare the arms in $D_m$. After the exploration phase, each player forms local empirical estimates of feasible actions using only their own observed rewards and then commits to their empirically best feasible action until the next epoch boundary.

**Theorem 12** *The regret of `Joint-mDSEE` satisfies*

$$R_T = O(\lfloor \log_2 T \rfloor \, f(\lfloor \log_2 T \rfloor)\, KM)\,.$$

The multiplicative factor $K$ reflects the additional exploration required to distinguish among the $K$ local arms of each player. As in the binary case, the dependence on the reward gaps is absorbed into the choice of the exploration schedule $f$. For any fixed set of gaps, the regret remains logarithmic in $T$. The proof is deferred to the appendix.

## 6 Experimental Results

We validate our theoretical bounds through experiments on both $0-1$ and joint arm combinatorial bandits, comparing `mCombinatorialUCB-A` (action asymmetry), `mCombinatorialUCB-B` (reward asymmetry with signaling), and `mDSEE` (full asymmetry).

**Setup** We consider $M$ players where each player has $K$ arms plus a null action (0). The total action space contains $(K+1)^M - 1$ joint actions (excluding the all-zero action). The feasible set $\mathcal{F}$ is a random subset of this space, with $|\mathcal{F}|$ denoting its cardinality. Per-player true arm rewards are drawn i.i.d. from $\mathrm{Unif}[0.2, 0.8]$, and stochastic rewards follow $\mathrm{Beta}(\alpha, \beta)$ distributions with concentration $\kappa = 10$, where $\alpha = \mu\kappa$ and $\beta = (1-\mu)\kappa$ for mean $\mu$.[1] We also set $f(x) = x$. We report averages over 10 independent runs with $\pm 1$ standard deviation bands.

**Binary Combinatorial Bandits** With $M = 5$, $K = 1$, $|\mathcal{F}| = 19$ (60% of 31 possible actions), and $T = 150{,}000$ (Figure 1), `mCombinatorialUCB-A` attains lowest cumulative regret ($\approx$15k), leveraging synchronized

---

[1]Note that any value drawn from Beta is in $[0,1]$ which is $(1/4)$-subGaussian by Hoeffding's Lemma. See detailed proof in Appendix.

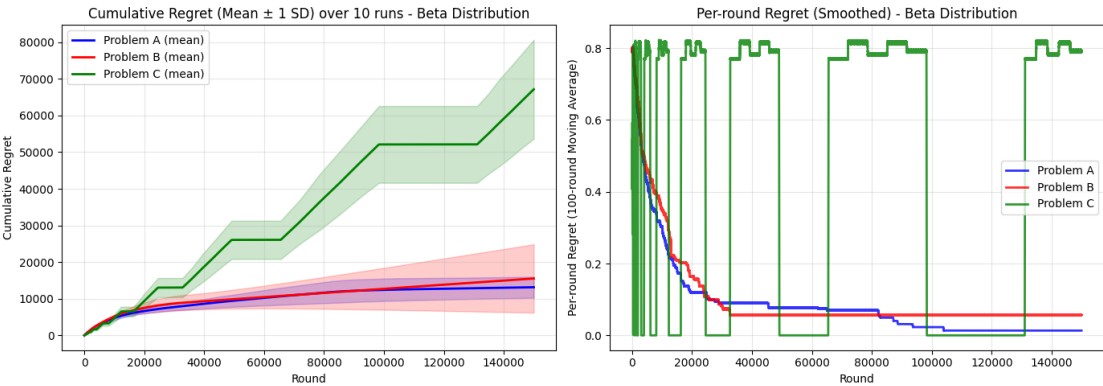

Figure 1: Binary setting: $M = 5$, $K = 1$, $|\mathcal{F}| = 19$, $T = 150{,}000$.

confidence bounds. `mCombinatorialUCB-B` shows higher variance due to signaling events that temporarily eliminate feasible actions, consistent with the additive $|\mathcal{F}|$ term in Theorem 5. `mDSEE` displays periodic exploration phases at each power of 2, creating its characteristic staircase pattern in cumulative regret with 0 regret during later commit phases.

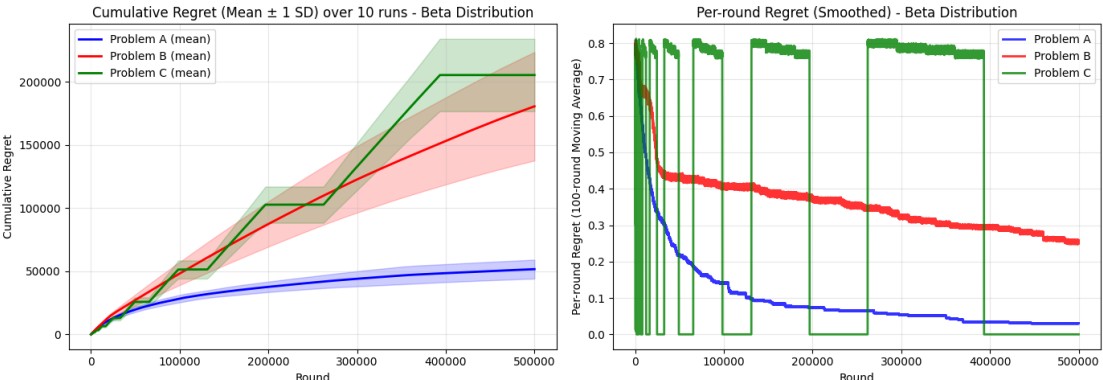

Figure 2: Multi-arm setting: $M = 4$, $K = 3$, $|\mathcal{F}| = 51$, $T = 500{,}000$.

**Multi-Arm Combinatorial Bandits** With $M = 4$, $K = 3$, $|\mathcal{F}| = 51$ (20% of 255 possible actions), and $T = 500{,}000$, cumulative regret increases roughly $K^2$-fold compared to the binary case (Figure 2). `mCombinatorialUCB-A` maintains its advantage ($\approx$50k regret), while `mCombinatorialUCB-B` reaches $\approx$180k with increased signaling complexity and more difficulty eliminating arms. `mDSEE` shows the steepest scaling ($\approx$220k), requiring $K$ times more exploration per epoch as predicted by the $KM$ factor in Theorem 7.

**Code Reproducibility** Complete implementation and experimental code to reproduce all results are available at `https://anonymous.4open.science/status/combinatorial_bandits_code-FE60`.

## 7 Conclusions and Future Work

We studied combinatorial bandits under multiplayer information asymmetry and showed that coordination without communication is possible with only polynomial overhead in the number of players and arms. Using pre-agreed exploration orders (Problem A) or observable deviations (Problem B), we achieve near-optimal $\tilde{\mathcal{O}}(\sqrt{T})$ regret, matching single-player rates up to problem-dependent factors.

Moving from single-arm ($K = 1$) to multi-arm selection ($K > 1$) introduces only a $K^2$ multiplicative regret factor, indicating that coordination—not combinatorial complexity—is the primary challenge. Our gap-independent bounds also eliminate the exponential dependence on $1/\Delta_{\min}^2$ present in prior work.

Open questions include the tightness of the $M^{3/2}$ dependence, whether signaling-based coordination in Problem B is inherently costly (particularly when $|\mathcal{F}| = 2^M$, where the additive $|\mathcal{F}|$ term dominates the bound; see Remark 6), and whether Problem C admits gap-independent polylogarithmic regret without the exploration function. A further question is when the general combinatorial feasibility structure genuinely requires our coordinated algorithms versus when the problem decomposes into independent per-player sub-problems (as is the case for $|\mathcal{F}| = \{0, 1\}^M$). Extensions to time-varying feasible sets or intermediate observability models are promising future directions.

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

## A  Beta Distributions are 1-subGaussian via Hoeffding's Lemma

**Definition 13 ($\sigma^2$-subGaussian)** *A centered random variable $X$ is $\sigma^2$-subGaussian if*

$$\mathbb{E}\, e^{\lambda(X-\mathbb{E}X)} \leq \exp\!\left(\tfrac{\lambda^2 \sigma^2}{2}\right) \quad \textit{for all } \lambda \in \mathbb{R}.$$

*Equivalently, $X$ has tails dominated by a Gaussian with variance proxy $\sigma^2$.*

**Theorem 14 (Hoeffding's Lemma)** *Let $X$ be any real-valued random variable with $a \leq X \leq b$ almost surely. Then*

$$\mathbb{E}\, e^{\lambda(X-\mathbb{E}X)} \leq \exp\!\left(\tfrac{\lambda^2(b-a)^2}{8}\right) \quad \textit{for all } \lambda \in \mathbb{R}.$$

In particular, $X - \mathbb{E}X$ is sub-Gaussian with variance proxy $\sigma^2 = (b-a)^2/4$; see (Boucheron et al., 2013, Ch. 2).

**Proposition 15 (Beta is 1-subGaussian)** *If $X \sim \mathrm{Beta}(\alpha, \beta)$, then $X - \mathbb{E}X$ is sub-Gaussian with variance proxy $\sigma^2 \leq \frac{1}{4}$.*

**Proof:**  If $X \sim \mathrm{Beta}(\alpha, \beta)$ with $\alpha, \beta > 0$, then $X$ is supported on the interval $[0, 1]$.

Applying Theorem 14 with $(a, b) = (0, 1)$ gives

$$\mathbb{E}\, e^{\lambda(X-\mathbb{E}X)} \leq \exp\!\left(\tfrac{\lambda^2}{8}\right) = \exp\!\left(\tfrac{\lambda^2}{2} \cdot \tfrac{1}{4}\right),$$

so $X - \mathbb{E}X$ is $\frac{1}{4}$-subGaussian. Since sub-Gaussianity is monotone in the variance proxy, $X - \mathbb{E}X$ is also 1-subGaussian. $\qquad\square$

## B  Important Lemmas

We present the proof of Lemma 2.

**Proof:**  By De Morgan's law,

$$G^c = \left(\bigcap_{t=1}^{T}\bigcap_{a=1}^{M} G_a(t)\right)^c = \bigcup_{t=1}^{T}\bigcup_{a=1}^{M} G_a(t)^c. \tag{9}$$

Thus, at least one arm at some time violates the confidence bound. For each arm $a$ and round $t$,

$$G_a(t)^c = \left\{ |\widehat{\mu}_a(t-1) - \mu_a| > \sqrt{\frac{4\log T}{n_a(t-1)}} \right\}.$$

Using the concentration bound with $\delta = T^{-2}$ gives

$$\Pr(G_a(t)^c) \leq \frac{1}{T^2}.$$

Applying the union bound,

$$\Pr(G^c) = \Pr\left(\bigcup_{t=1}^{T}\bigcup_{a=1}^{M} G_a(t)^c\right) \leq \sum_{t=1}^{T}\sum_{a=1}^{M} \Pr(G_a(t)^c).$$

Hence

$$\Pr(G^c) \leq T \cdot M \cdot \frac{1}{T^2} = \frac{M}{T}. \tag{10}$$

Therefore,

$$\Pr(G^c) \leq \min\!\left(\frac{M}{T}, 1\right).$$

$$\square$$

## C    Proofs for the 0–1 Problem

### C.1    The Interval Approach for the 0–1 Problem (Problem A and Problem B)

We present the proof of Theorem 4.

**Proof:**    Let $\mu^\star = \mu_{\vec{a}^\star}$ denote the mean reward of the optimal feasible action, and let $X_{\vec{a}_t}$ be the reward collected at round $t$.

$$R_T \leq \sum_{t=1}^{T} \mathbb{E}[\mu^\star - X_{\vec{a}_t}] \tag{11}$$

$$= \sum_{t=1}^{T} \mathbb{E}[\mu^\star - X_{\vec{a}_t} \mid G]P(G) + \sum_{t=1}^{T} \mathbb{E}[\mu^\star - X_{\vec{a}_t} \mid G^c]P(G^c) \tag{12}$$

$$= P(G) \sum_{t=1}^{T} \mathbb{E}[\mu^\star - X_{\vec{a}_t} \mid G] + P(G^c) \sum_{t=1}^{T} \mathbb{E}[\mu^\star - X_{\vec{a}_t} \mid G^c] \tag{13}$$

$$\leq \sum_{t=1}^{T} \mathbb{E}[\mu^\star - \mu_{\vec{a}_t} \mid G] + M. \tag{14}$$

Here $G$ is the good event from (5). The last inequality uses: (i) $P(G) \leq 1$, (ii) $P(G^c) \leq M/T$ by Lemma 2, and (iii) per-round regret is at most 1.

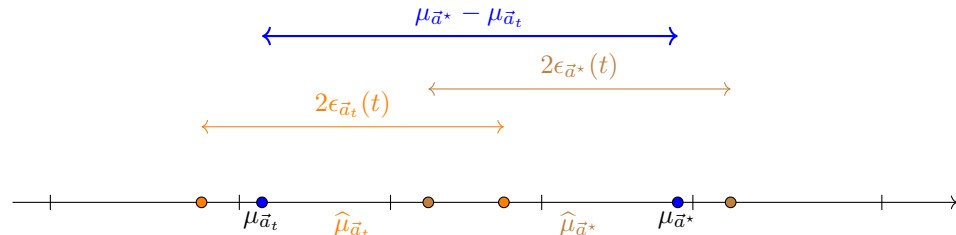

Denote the confidence interval for $\mu_{\vec{a}}$ by $I_{\vec{a}}(t)$ as in (4). At round $t$, the played action $\vec{a}_t$ is not dominated by $\vec{a}^\star$, so their intervals overlap under the good event. Hence

$$\mu_{\vec{a}_t} \in I_{\vec{a}_t}(t), \qquad \mu^\star \in I_{\vec{a}^\star}(t),$$

which implies

$$\mathbb{E}[\mu^\star - X_{\vec{a}_t} \mid G] \leq 2\epsilon_{\vec{a}_t}(t) + 2\epsilon_{\vec{a}^\star}(t). \tag{15}$$

We now bound the two sums carefully. Note that $\epsilon_a(t) = \sqrt{4\log T / n_a(t)}$ depends on the individual pull count $n_a(t)$, *not* on the round index $t$.

**First term.**    For each arm $a \in [M]$, summing $\epsilon_a(t)$ only over the rounds in which arm $a$ is pulled gives the standard integral bound

$$\sum_{t:\, a \in \vec{a}_t} \epsilon_a(t) = \sum_{s=1}^{n_a(T)} \sqrt{\frac{4\log T}{s}} \leq 2\sqrt{4\, n_a(T)\log T}.$$

Summing over arms and applying Cauchy–Schwarz with $\sum_{a=1}^{M} n_a(T) \leq MT$,

$$\sum_{t=1}^{T} 2\epsilon_{\vec{a}_t}(t) = \sum_{a=1}^{M} 2 \sum_{t:\, a \in \vec{a}_t} \epsilon_a(t) \leq 4\sqrt{\log T} \sum_{a=1}^{M} \sqrt{n_a(T)} \leq 4\sqrt{\log T} \cdot \sqrt{M} \cdot \sqrt{MT} = 4M\sqrt{T\log T}.$$

**Second term.** For the optimal action $\vec{a}^{\star}$, each component $a \in \vec{a}^{\star}$ is fixed across all rounds. Because the algorithm cycles through $S_1, \ldots, S_M$, each such component is pulled at least once every $M$ rounds, so $n_a(t) \geq \lfloor t/M \rfloor$ and hence

$$\epsilon_a(t) = \sqrt{\frac{4 \log T}{n_a(t)}} \leq \sqrt{\frac{4M \log T}{t}}.$$

Therefore

$$\sum_{t=1}^{T} \epsilon_a(t) \leq \sqrt{4M \log T} \sum_{t=1}^{T} \frac{1}{\sqrt{t}} \leq \sqrt{4M \log T} \cdot 2\sqrt{T} = 4\sqrt{MT \log T}.$$

Summing over the (at most $M$) arms in $\vec{a}^{\star}$,

$$\sum_{t=1}^{T} 2\epsilon_{\vec{a}^{\star}}(t) \leq 2M \cdot 4\sqrt{MT \log T} = 8M^{3/2}\sqrt{T \log T}.$$

Substituting into (15) yields

$$R_T \leq 4M\sqrt{T \log T} + 8M^{3/2}\sqrt{T \log T} + M \leq 12\, M^{3/2}\sqrt{T \log T} + M.$$

$\square$

## C.2 `mDSEE` for Problem C

We present the proof of Theorem 7.

**Proof:** We divide the regret into exploration and commitment phases. During exploration,

$$R_{\text{explore}} \leq \lfloor \log_2(T) \rfloor f(\lfloor \log_2(T) \rfloor) M.$$

For each feasible action $\vec{a} \in \mathcal{F}$, define

$$G_{\vec{a}}(t) = \left\{ |\widehat{\mu}_{\vec{a}} - \mu_{\vec{a}}| < \epsilon \right\}, \qquad \epsilon = \frac{1}{2} \min_{\vec{a} \in \mathcal{F}} \Delta_{\vec{a}}.$$

For the commitment phase,

$$R_{\text{commit}} \leq \sum_{t=1}^{T} \mathbb{E}\left[ \mathbb{I}\left\{ \left( \bigcap_{\vec{a} \in \mathcal{F}} G_{\vec{a}}(t) \right)^c \right\} \right] \tag{16}$$

$$= \sum_{t=1}^{T} \Pr\left( \bigcup_{\vec{a} \in \mathcal{F}} G_{\vec{a}}(t)^c \right) \tag{17}$$

$$\leq \sum_{t=1}^{T} \sum_{\vec{a} \in \mathcal{F}} \Pr(G_{\vec{a}}(t)^c) \tag{18}$$

$$\leq \sum_{t=1}^{T} \sum_{\vec{a} \in \mathcal{F}} e^{-n_{\vec{a}}(T)\epsilon^2/2} \tag{19}$$

$$\leq \sum_{t=1}^{T} M e^{-M_0(t) \log_2(t)\epsilon^2/2} \tag{20}$$

$$= \sum_{t=1}^{T} M t^{-M_0(t)\epsilon^2/(2\log 2)}. \tag{21}$$

Here $M_0(t)$ is a lower bound on the number of times each base arm in the support of any feasible action has been sampled during exploration. Since $\log_2(t)$ grows with $t$, this term decays super-polynomially. Hence

$$R_{\text{commit}} \leq \sum_{t=1}^{T} Mt^{-c} = O(1)$$

for some constant $c > 0$. Therefore,

$$\begin{aligned} R_T &\leq R_{\text{explore}} + R_{\text{commit}} \\ &\leq \lfloor \log_2(T) \rfloor f(\lfloor \log_2(T) \rfloor)M + O(1) \\ &= O(\lfloor \log_2(T) \rfloor f(\lfloor \log_2(T) \rfloor)M)\,. \end{aligned}$$

$\square$

## D  The Joint Arm Problem Pseudocode

The pseudocode for Joint Arm Problem A is given in Algorithm 4.

---

**Algorithm 4** `mCombinatorialUCB-Joint-A`

---

**Require:** Horizon $T$; player sets $S_1, \ldots, S_M$ with fixed internal orders; per-player choice lists $D_1, \ldots, D_M$
**Ensure:** Cyclic elimination and UCB updates for individual and combinatorial arms
1: **Initialize:** for each individual arm $a_i^j$, set $n_{a_i^j}(0) \leftarrow 0$, $\hat{\mu}_{a_i^j}(0) \leftarrow 0$, $\text{UCB}_{a_i^j}(0) \leftarrow +\infty$, $\text{LCB}_{a_i^j}(0) \leftarrow -\infty$
2: $i \leftarrow 1$
3: **for** $t = 1$ **to** $T$ **do**
4:     Let $S_i$ be the current set and let $\vec{a}$ be the next combinatorial arm in (6)
5:     Let $a_i^k$ be the current arm in $D_i$
6:     $\vec{a}_t \leftarrow \vec{a}$
7:     **while exists** $a' \in D_i$ **such that** $\text{UCB}_{a_i^k}(t-1) < \text{LCB}_{a'}(t-1)$ **do**
8:         Remove $a_i^k$ from $D_i$
9:         Set $a_i^k$ to the next element of $D_i$
10:    **end while**
11:    **while exists** $\vec{a}' \in S_i$ **such that** $\text{UCB}_{\vec{a}_t}(t-1) < \text{LCB}_{\vec{a}'}(t-1)$ **do**
12:        Remove $\vec{a}_t$ from all $S_j$ that contain it
13:        Set $\vec{a}_t$ to the next arm in $S_i$
14:    **end while**
15:    **for all** $a_i^j \in \vec{a}_t$ **do**
16:        Observe reward $X_{a_i^j}(t)$
17:        Update $n_{a_i^j}(t)$, $\hat{\mu}_{a_i^j}(t)$, $\text{UCB}_{a_i^j}(t)$, $\text{LCB}_{a_i^j}(t)$
18:    **end for**
19:    **for all** $\vec{b} \in S_i$ **do**
20:        Update $\text{UCB}_{\vec{b}}(t)$ and $\text{LCB}_{\vec{b}}(t)$
21:    **end for**
22:    $i \leftarrow (i \bmod M) + 1$
23: **end for**

---

The pseudocode for Joint Arm Problem B is given in Algorithm 5.

The pseudocode for Joint Arm Problem C is given in Algorithm 6.

### D.1  Proofs for the Interval Approach for Problem A and Problem B

We present the proof of Theorem 10.

---

**Algorithm 5** `mCombinatorialUCB-Joint-B`

---

**Require:** Horizon $T$; player sets $S_1, \ldots, S_M$ with fixed internal orders; per-player choice lists $D_1, \ldots, D_M$

**Ensure:** Joint elimination via player-triggered sabotage and UCB updates

1: **Initialize:** for each individual arm $a_i^j$, set $n_{a_i^j}(0) \leftarrow 0$, $\hat{\mu}_{a_i^j}(0) \leftarrow 0$, $\text{UCB}_{a_i^j}(0) \leftarrow +\infty$, $\text{LCB}_{a_i^j}(0) \leftarrow -\infty$

2: $i \leftarrow 1$

3: **for** $t = 1$ **to** $T$ **do**

4:     Let $S_i$ be the current set and let $\vec{a}$ be the next combinatorial arm in (6)

5:     Let $a_i^k$ be the current arm in $D_i$

6:     $\vec{a}_t \leftarrow \vec{a}$

7:     **if exists** $a' \in D_i$ **such that** $\text{UCB}_{a_i^k}(t-1) < \text{LCB}_{a'}(t-1)$ **then**

8:         Player $i$ deviates to signal elimination of $a_i^k$

9:         All players remove $a_i^k$ from $D_i$

10:         Set $a_i^k$ to the next element of $D_i$

11:     **end if**

12:     **if exists** player $j \neq i$ and **exists** $\vec{a}' \in S_i$ **such that** $\text{UCB}_{\vec{a}_t}(t-1) < \text{LCB}_{\vec{a}'}(t-1)$ **then**

13:         Player $j$ deviates so that the resulting joint action is not $\vec{a}_t$

14:         All players remove $\vec{a}_t$ from every $S_r$ that contains it

15:         Set $\vec{a}_t$ to the next arm in $S_i$

16:     **end if**

17:     **for all** $a_i^j \in \vec{a}_t$ **do**

18:         Observe reward $X_{a_i^j}(t)$

19:         Update $n_{a_i^j}(t)$, $\hat{\mu}_{a_i^j}(t)$, $\text{UCB}_{a_i^j}(t)$, $\text{LCB}_{a_i^j}(t)$

20:     **end for**

21:     **for all** $\vec{b} \in S_i$ **do**

22:         Update $\text{UCB}_{\vec{b}}(t)$ and $\text{LCB}_{\vec{b}}(t)$

23:     **end for**

24:     $i \leftarrow \left(i \bmod M\right) + 1$

25: **end for**

---

---

**Algorithm 6** `JointArm-mDSEE` for Problem C

---

**Require:** Number of players $M$, number of individual arms $K$, feasible set $\mathcal{F}$, monotone function $f(\lambda)$ with
   $f(\lambda) \to \infty$
**Ensure:** Identification of high-reward joint arms
 1: $t \leftarrow 1$, $\lambda \leftarrow 1$
 2: **while** $t \leq T$ **do**
 3:   **Exploration phase:**
 4:   **for** $m = 1$ **to** $M$ **do**
 5:     **for** $k = 1$ **to** $K$ **do**
 6:       **for** $p = 1$ **to** $M$ **do**
 7:         Pull the representative joint arm $\vec{a} \in S_p$
 8:         **if** $p = m$ **then**
 9:           Pull arm $a_k^p$ from $D_p$
10:         **else**
11:           Pull the arm in $D_p$ with largest empirical mean
12:         **end if**
13:         Observe reward $X_{a_k^p}^{(p)}(t)$
14:       **end for**
15:       Update empirical means for player $m$
16:     **end for**
17:   **end for**
18:   **Commit phase:**
19:   **for** $m = 1$ **to** $M$ **do**
20:     Compute local empirical means for all $\vec{a} \in \mathcal{F}$
21:     $\vec{a}_{\text{commit}}^m \leftarrow \arg\max_{\vec{a} \in \mathcal{F}} \hat{\mu}_{\vec{a}}^m$
22:   **end for**
23:   $\lambda \leftarrow \lambda + 1$, $t \leftarrow t + 1$
24: **end while**

---

**Proof:** Recall that the upper endpoint of the interval for a combinatorial action is obtained by maximizing over the surviving individual arms for each player. Let this maximizing action be denoted by $\vec{a}_t^\star$. Then under the good event,

$$\mathbb{E}[\mu^\star - X_{\vec{a}_t} \mid G] \leq 2 \sum_{a \in \vec{a}_t} \epsilon_a(t) + 2 \sum_{a \in \vec{a}_t^\star} \epsilon_a(t) + 2 \sum_{a \in \vec{a}^\star} \epsilon_a(t).$$

In the joint-arm setting, the cyclic ordering combined with the two-level elimination cycles through at most $KM$ individual (player, arm) pairs before returning to the same pair. Consequently, for any individual arm $a \in \vec{a}^\star$ that is fixed across rounds, $n_a(t) \geq t/(KM)$ and $\epsilon_a(t) \leq \sqrt{4KM \log T/t}$.

For the first term, summing over the rounds in which each individual arm is actually pulled and applying Cauchy–Schwarz,

$$2 \sum_{t=1}^{T} \sum_{a \in \vec{a}_t} \epsilon_a(t) \leq 4\sqrt{\log T} \sum_{a} \sqrt{n_a(T)} \leq 4\sqrt{KM \cdot T \log T} \cdot \sqrt{KM} = 4KM\sqrt{T \log T}.$$

For the second term, the same analysis applied to the $K$ currently surviving arms per player in $\vec{a}_t^\star$ contributes an extra factor of at most $K$:

$$2 \sum_{t=1}^{T} \sum_{a \in \vec{a}_t^\star} \epsilon_a(t) \leq 4K^2 M\sqrt{T \log T}.$$

For the third term, the components of the fixed optimal action $\vec{a}^\star$ satisfy $n_a(t) \geq t/(KM)$, so

$$2 \sum_{t=1}^{T} \sum_{a \in \vec{a}^\star} \epsilon_a(t) \leq 2M \cdot \sqrt{4KM \log T} \cdot 2\sqrt{T} = 8\,K^{1/2}M^{3/2}\sqrt{T \log T} \leq 8\,K^2 M^{3/2}\sqrt{T \log T}.$$

Combining the three terms gives the stated bound up to constants. $\qquad\square$

### D.2    Joint `mDSEE` algorithm proof

We present the proof of Theorem 12.

**Proof:** As before, we split regret into exploration and commitment phases. The exploration phase contributes

$$R_{\text{explore}} \leq \lfloor \log_2(T) \rfloor f(\lfloor \log_2(T) \rfloor) KM.$$

Define

$$G_{\vec{a}}(t) = \{|\widehat{\mu}_{\vec{a}} - \mu_{\vec{a}}| < \epsilon\}, \qquad \epsilon = \frac{1}{2} \min_{\vec{a} \in \mathcal{F}} \Delta_{\vec{a}}.$$

Then

$$R_{\text{commit}} \leq \sum_{t=1}^{T} \mathbb{E}\left[ \mathbb{I}\left\{ \left( \bigcap_{\vec{a} \in \mathcal{F}} G_{\vec{a}}(t) \right)^c \right\} \right] \tag{22}$$

$$= \sum_{t=1}^{T} \Pr\left( \bigcup_{\vec{a} \in \mathcal{F}} G_{\vec{a}}(t)^c \right) \tag{23}$$

$$\leq \sum_{t=1}^{T} \sum_{\vec{a} \in \mathcal{F}} \Pr(G_{\vec{a}}(t)^c) \tag{24}$$

$$\leq \sum_{t=1}^{T} M e^{-M_0(t) \log_2(t) \epsilon^2/2}. \tag{25}$$

Thus $R_{\text{commit}} = O(1)$, and therefore

$$R_T \leq R_{\text{explore}} + R_{\text{commit}} = O(\lfloor \log_2(T) \rfloor f(\lfloor \log_2(T) \rfloor) KM).$$

$\qquad\square$

