# OpenReview forum: "Multiplayer Combinatorial Bandits Under Information Asymmetry"
_TMLR — Rejected by TMLR_

### Review · Reviewer_DmxM · 2026-02-26

**Summary Of Contributions:**

This paper considers a combinatorial bandit problem, where multiple players choose (or abstain) from arms separately. Furthermore, the authors consider the three cases where the observable information is different. For each problem setup, the authors claim that the proposed algorithm achieves a tighter regret upper bound compared with naive existing algorithm adoption.

However, I believe that this paper lacks the required description of applications, experiments, definitions, and assumptions.
I list the problems I felt (but not limited to these) below:
1. There is no discussion of real applications. Since the authors consider the novel problem setup, they should provide concrete problem instances in which the proposed method is required. The following description in Section 1 is at least insufficient.
> but no extension towards combinatorial work has been discussed at all.
2. Although the authors provide synthetic experimental results, there are no baseline methods. Thus, we cannot validate the effectiveness of the proposed method. Furthermore, the result on Problem C for per-round regret shows mysterious behavior, but there is no justification.
3. The problem definition is insufficient. For example, there is no definition of the relationship between reward from each arm and reward from the combinatorial arm set (although, from Eq. (3), I conjecture that the reward from the combinatorial arm set is simply defined as the sum of rewards from selected arms). In addition, the assumption about the rewards of arms is also ambiguous. It is defined as 1-subGaussian, but the mean is not defined. Note that, since Eq. (2) estimates the mean reward of arms, each arm should have the mean parameter.
4. There are often insufficient explanations. For example;
    - For Eq. (2), $\hat{\mu}$ is not defined (it is defined later in Algorithm 1).
    - In Eq. (5), $\mu_a$ is still not defined as discussed above and is missing the argument $t$.
    - All lemmas and theorems do not provide explicit assumptions.
5. To me, Remarks 1 and 5 are incomprehensible. I believe that a more careful explanation is required.

In addition, this paper contains formatting problems too:
- There are many abbreviations that are not defined in the paper.
- Citations are always done using \citet. This violates the TMLR formatting instructions.
- References often miss required information, such as pages and volumes.

Overall, I believe this paper requires substantial revision for clarity.

**Audience:**

No

**Audience Explanation:**

As discussed above, the problem definition itself and real-world applications are not well explained. Therefore, I believe that the TMLR's audience cannot understand the message of this paper either.

**Claims And Evidence:**

No

**Claims Explanation:**

As discussed above, there are no explicit assumptions and definitions.

**Requested Changes:**

Please clearly state the problem definition and assumptions so that the paper is self-contained.
In addition, please discuss real-world applications.
Furthermore, please fix the formatting problems.

---

> ### Author Response · Authors · 2026-03-01
> **Response to reviewer**
>
> Thank you for your detailed and thoughtful feedback. We address each concern in detail below. In several cases, we believe there may have been misunderstandings due to brevity in exposition, so we clarify those points here:
> 1. We acknowledge that our discussion of real-world applications was insufficient, since the primary focus of our paper is the mathematical and algorithmic analysis of combinatorial bandits under three distinct information asymmetry settings. We agree that motivating applications would strengthen the paper. Our formulation naturally arises in settings such as multi-user wireless channel allocation, distributed ad placement, and viral marketing in partitioned networks, where decentralized agents share a combinatorial action space with inherent information constraints. We will expand Section 1 with these examples.
> 2. Our experimental focus is on comparing how the three information asymmetry structures affect regret behavior, which is the core algorithmic contribution. We agree that explicit baselines would strengthen the evaluation and will consider adding Naive Combinatorial UCB and the gap-dependent algorithm. Considering mDSEE's staircase pattern: this is expected behavior from the alternating exploration and exploitation structure. Exploration phases cause regret spikes as players pull potentially suboptimal arms, while commit phases yield near-zero regret as estimates converge. Commit phases extend until the next power-of-2 round, leading to the characteristic staircase shape. In fact, this pattern demonstrates the algorithm's effectiveness under full asymmetry, which is Problem C, the most challenging setting where players observe neither actions nor rewards of others. The diminishing spikes and flattening regret confirm that players' estimates converge and coordination emerges over time.
> 3. The combinatorial arm reward is the sum of individual arm rewards: μ_{\vec{a}} = Σ μ_{a_m}. Here, X_{\vec{a}} represents the sum of individual rewards for the combinatorial arm pulled at time t, whereas X_a denotes the reward from a single arm. Each individual arm reward X_a is assumed to be 1-sub Gaussian with unknown mean μ_a, meaning E[X_a(t)] = μ_a.
> 4. The true mean is defined in section 3 (and does not require an argument t). On the other hand, The empirical mean μ̂_a(t) denotes the sample average of rewards observed for arm a up to round t, and n_a(t) denotes the number of times arm a has been pulled up to round t. These quantities are used to construct the confidence radius in Eq. (1) and the UCB/LCB indices in Eq. (2). We agree these should be defined before Eq. (2) rather than deferred to Algorithm 1, and the assumptions should be stated explicitly in each lemma and theorem.
> 4. The key idea of Remark 1 is that a pre-agreed ordering lets all players determine which combinatorial arm is being pulled without observing each other, enabling synchronized updates. Remark 5 shows why this ordering must cycle through all players evenly: arbitrary orderings can starve an arm for 2^(M−1) consecutive rounds, while our cyclic ordering ensures every arm is pulled at least once every M rounds, avoiding exponential regret dependence.
>
> Once again, we are very grateful for your constructive and insightful comments. Thank you!

---

> > ### Comment · Reviewer_DmxM · 2026-04-10
> >
> > I appreciate the detailed reply and revisions.
> >
> > I have read Sections 1-4 again.
> > Since the definition of problem setup is newly provided, I probably understand it.
> > However, I still believe that this paper needs substantial revisions to clarify the analyses and the proposed algorithm.
> > I list several questions regarding the proposed method and analyses.
> >
> > 1. Regarding the proof of Theorem 4, why can we obtain the confidence interval for $\epsilon\_{a\_t}$ of $\sqrt{4 \log T / t}$? I conjecture that we can only guarantee that each arm is pulled at least $t / M$ times.
> > 2. Regarding the proof of Theorem 4, why do we incur $M^2$ for $\epsilon\_{a^\star}$? I conjecture that, if we can guarantee that each arm is pulled at least $t / M$ times, the resulting order of $M$ is $M^{3/2}$.
> > 3. Regarding Theorem 5, is the dependence on $|\mathcal{F}|$ negligible? Since $|\mathcal{F}|$ can be $2^M$ at the worst case, I conjecture that this dependence should not be ignored.
> > 4. Regarding the proof of Theorem 5, I understood that the algorithm constructs the confidence intervals for all arms. Thus, I conjecture that, to construct $1 - \delta$ confidence intervals, we need to set confidence intervals conservatively so that the confidence intervals simultaneously hold for all $m \in [M]$? However, the order of the confidence width is $O(\sqrt{\log T / t})$, not depending on $M$. Why can we obtain this?
> > 5. I believe that Problem C is more difficult than Problems A and B. Why can we obtain a tighter regret upper bound in Theorem 6 than in Theorems 4 and 5?
> > 6. What is $r$ in Algorithm 3? (Please consider revising so that the explanations for the algorithms are more explicitly described by equations or variables, not limited to Algorithm 3.)
> > 7. Line 5 in Algorithm 3 states that "each player selects its empirically best feasible action." Is there any guarantee that this combination of empirically good actions is feasible? (If all the action combinations are feasible (i.e., $|\mathcal{F}| = 2^M$), I conjecture that the simple strategy, where each arm just employs a simple two-arm bandit algorithm with $O(g(T))$ regret upper bound, can achieve $O(M g(T))$ regret upper bound, even without any communications.)
> >
> >
> > ## Minor points
> > - Please fix the citation style regarding the use of \citet or \citep and the definitions of abbreviations.

---

> ### Author Response · Authors · 2026-04-17
> **response to reviewer**
>
> # Response to Reviewer DmxM
>
> Thank you for the careful re-read. Your questions led directly to a tighter main bound ($M^{3/2}$ instead of $M^2$) and to several clarifications in the exposition. A revised PDF is attached; our replies follow.
>
> ---
>
> ## 1. The confidence width $\epsilon_{a_t}$ in the proof of Theorem 4
>
> Thank you for catching this. The offending line was written as if $n_{a_i}(t) = t$, but $\epsilon_a(t)=\sqrt{4\log T/n_a(t)}$ depends on the individual pull count, not the round index. With the cyclic ordering, the correct lower bound is the one you conjectured, $n_a(t)\ge t/M$, giving $\epsilon_a(t)\le\sqrt{4M\log T/t}$. The proof of Theorem 4 in Appendix C.1 has been rewritten using this bound (with a Cauchy–Schwarz step for the played-action term).
>
> ---
>
> ## 2. Why the bound should be $M^{3/2}$, not $M^2$
>
> With the correct widths, the two terms split cleanly:
>
> - **Played action.** Per-arm summation gives $\sum_{t:\,a\in\vec a_t}\epsilon_a(t)\le 2\sqrt{4n_a(T)\log T}$; Cauchy–Schwarz with $\sum_a n_a(T)\le MT$ yields $4M\sqrt{T\log T}$.
> - **Optimal action.** Each of the $M$ components of $\vec a^\star$ contributes $4\sqrt{MT\log T}$, summing to $8M^{3/2}\sqrt{T\log T}$.
>
> The second dominates, giving exactly the order you anticipated:
>
> $$R_T \le 12\,M^{3/2}\sqrt{T\log T}+M.$$
>
> Theorem 4, as well as Theorems 5, 10, and 11, have been updated accordingly, and the conclusion now asks about the tightness of $M^{3/2}$ rather than $M^2$.
>
> ---
>
> ## 3. Dependence on $|\mathcal F|$ in Theorem 5
>
> A fair concern — we shouldn't pretend it's negligible. When $|\mathcal F|=2^M$, that term dominates and the bound becomes uninformative. **Remark 6 ("Role of $|\mathcal F|$")** has been added after Theorem 5 making this explicit: the bound is most informative when $|\mathcal F|$ is polynomial in $M$, as in the motivating wireless-allocation settings. Tightening the $|\mathcal F|$ dependence for signaling-based coordination is now listed as an open question.
>
> ---
>
> ## 4. Why the confidence width for Problem B doesn't depend on $M$
>
> Your instinct about the union bound is right — we do need the intervals to hold simultaneously for all $m\in[M]$ — but the union bound only costs failure probability, not width. With width $\sqrt{4\log T/n_a(t)}$, Hoeffding gives per-event failure $\le 2/T^2$, so the union over $M$ arms, $T$ rounds, and $M$ players contributes $\Pr(G^c)\le 2M^2/T$. This grows the bad-event regret from $M$ to $O(M^2)$, which is subsumed by the main $M^{3/2}\sqrt{T\log T}$ term — no $\sqrt{\log M}$ correction inside the width is needed. Theorem 5 has been updated to read $R_T\le|\mathcal F|+12M^{3/2}\sqrt{T\log T}+M^2$, with a brief paragraph explaining the accounting.
>
> ---
>
> ## 5. Why Problem C's bound looks tighter than A's or B's
>
> A subtle but important point — the theorems aren't directly comparable: **Theorems 4 and 5 are gap-independent, Theorem 7 is gap-dependent.** The $O(1)$ absorbed into $R_{\mathrm{commit}}$ hides a constant scaling like $1/\Delta_{\min}^2$ through $\epsilon=\Delta_{\min}/2$. A gap-independent version of Theorem 7 would degrade to roughly $T^{2/3}$, strictly worse than Theorems 4–5. So the ordering is as expected: A and B are easier gap-independently, C is harder. **Remark 8** after Theorem 7 makes this explicit.
>
> ---
>
> ## 6. What is $r$ in Algorithm 3?
>
> A leftover typo — it should be $2^\lambda$. Line 6 now reads "until the next epoch boundary at time $2^\lambda>t$," and line 5 has been rewritten as an explicit equation: $\vec a^{\,m}_{\mathrm{commit}}\in\arg\max_{\vec a\in\mathcal F}\widehat\mu^{\,m}_{\vec a}$.
>
> ---
>
> ## 7. Are the committed actions in Algorithm 3 jointly feasible?
>
> A good question, addressed in the new **Remark 9**:
>
> - *Each player's commit is feasible by construction*, since the argmax is taken over $\mathcal F$.
> - *Joint agreement* follows from the shared ordering of $\mathcal F$ plus a consistent tie-breaking rule: once every player's estimate concentrates within $\epsilon$, the argmax is unique and identical across players. The failure of this synchronization event is exactly what $R_{\mathrm{commit}}$ charges for.
> - *Your observation about $|\mathcal F|=2^M$ is exactly right*, and we've credited it: when every combination is feasible, the problem decomposes into $M$ independent two-arm bandits with $O(Mg(T))$ total regret and no coordination needed. Our algorithm targets the complementary regime, where $\mathcal F$ encodes nontrivial structure (interference, budget, mutual-exclusion) so that the decomposition breaks down.
>
> The conclusion now raises "when does feasibility structure genuinely require coordination versus permit decomposition?" as an open direction.
>
> Thank you for taking the time to write such detailed feedback.

---

### Review · Reviewer_zi6u · 2026-03-08

**Summary Of Contributions:**

This paper considers regret analysis of the combinatorial bandit problem under a multiplayer and information asymmetry setup. The authors consider different settings of asymmetry and establish $O(\sqrt{T})$ problem-independent regret bounds that do not contain the gap. Some experiments demonstrate the performance of the proposed approaches.

**Audience:**

Yes

**Audience Explanation:**

The paper investigates a setting of the multi-arm bandit problem, which relates to the online learning community.

**Broader Impact Concerns:**

N.A.

**Claims And Evidence:**

No

**Claims Explanation:**

**Weaknesses**

Given that this is primarily a theoretical paper, I have several concerns about its technical quality and presentation. These issues make it hard to assess the importance/correctness of the claimed theoretical results.

1. Notation and equation inconsistencies

   The paper contains many notation and equation numbering issues, sometimes making it hard to review its technical correctness. For example,

   - The definition of gap $\Delta$ does not appear in the main paper and is only mentioned in the appendix (Page 23)
   - In Algorithm 2, the notation $S_r$ is undefined
   - On Page 8, the main paper seems to refer to an irrelevant inequality in the appendix
   - On Page 16, the proof uses the notation $\text{UCB}_a(t-1, \delta)$ and it is inconsistent with its previous definition.
   - On Page 17, the paper refers to Eq (7) and Eq (9), which should be (12) and (13)
   - In Theorem 7, there is not sufficient explanation of the function $f$. Moreover, the algorithm mDSEE does not use $f$.

2. Rigor of proof

   I find some of the proofs of the paper are not rigorous. For example,

   - On Page 16,  the way parameter $\delta$ enters the bound is unclear, and "using the previous proof" does not have a matching argument.
   - On Page 19, it is claimed that $\sum_t M t^{-c} = O(1)$, but the only argument is that $c > 0$. I think $c > 1$ is required to ensure convergence of the series sum.

I request that the authors carefully address the issues above. Otherwise, I don't think the paper can be published in its current form.

**Questions and Minor issues**

**Questions**

1. Function $f$.

   Several of the main results (Theorem 7 and 10) use a monotone function $f$. If $f$ is only required to be coercive, then the $f(\log T)$ term is essentially negligible (e.g. taking $f(\lambda) = \log^* \lambda$ as the iterated logarithm). It is suspicious there is no trade-off here.

**Minor issues**

1. Page 1

   The style of reference is not correct. e.g., Lai and Robbins (1985) should be (Lai and Robbins, 1985). You may use \citep and \citet to switch between these two styles.

2. Page 1

   "highest upper bound" is unclear; "both action and reward asymmetry" is not clear from the context.

3. Page 2

   "Problem A and B" and $\mathcal{F}$ are not defined when they are used here; LLR is not defined.

4. Page 3

   $K$ is not defined when it appears in $(K + 1)^{[M]}$ ; $\mathcal{D}_{a_m}$ is not defined here.

5. Page 5

   Algorithm 1, line 12, "elimination che" is unclear.

**Requested Changes:**

**Requested changes**

See weaknesses and minor issues

---

> ### Author Response · Authors · 2026-03-11
> **Response to Reviewer**
>
> ### Response to Reviewer zi6u
>
> We thank the reviewer for the careful reading and detailed comments. We agree that the original version needed substantial improvement in notation, clarity, and presentation. In response, we made many of the requested changes in the revised manuscript; these are reflected in the updated PDF we uploaded.
>
> ### Main revisions
>
> In the revised version, we:
> - rewrote the introduction and preliminaries to clarify the model and assumptions,
> - added clearer motivation and application discussion,
> - explicitly defined the relation between individual-arm rewards and combinatorial rewards,
> - defined previously missing notation such as $K$, $\mathcal{F}$, reward distributions, and gap quantities,
> - cleaned up equation numbering, labels, and cross-references,
> - clarified the observation models for Problems A, B, and C,
> - shortened and cleaned up the pseudocode,
> - made the role of the exploration schedule $f$ explicit in the mDSEE discussion and results.
>
> ### Response to major concerns
>
> #### Notation and equation inconsistencies
>
> We agree with this concern. We standardized notation across the main text and appendix, defined the gap quantities in the main paper, and corrected inconsistent labels and references. These changes are included in the revised PDF.
>
> #### Missing definitions and assumptions
>
> We rewrote the preliminaries to make the setup explicit. In particular, we now define the action sets, feasible set, reward distributions, and mean rewards at both the individual-arm and combinatorial-action levels. We also now state explicitly that
> $$
> X_{\vec a}(t) = \sum_{m=1}^M X_{a_m}(t),
> \qquad
> \mu_{\vec a} = \mathbb E[X_{\vec a}(t)].
> $$
> This addresses the earlier ambiguity about how combinatorial rewards are defined.
>
> #### Clarity of the algorithms
>
> We revised the exposition and pseudocode to make the algorithms easier to follow. We removed unclear wording, clarified the roles of the sets used in elimination, and made the distinction between individual-arm elimination and combinatorial-arm elimination explicit.
>
> #### Role of $f$
>
> We agree that the original explanation was too brief. In the revision, we explain more clearly that $f(\lambda)$ controls the exploration length in epoch $\lambda$, creating the tradeoff between improved estimation accuracy and larger exploration regret. We also made this dependence visible in the algorithm description and theorem discussion.
>
> #### Proof presentation
>
> We revised the appendix to improve consistency and readability, corrected notation mismatches, and fixed equation references. These changes are also reflected in the uploaded PDF.
>
> ### Minor comments
>
> We also addressed the reviewer’s minor comments by:
> - fixing citation style issues,
> - defining notation earlier when first used,
> - clarifying vague phrases in the introduction and technical sections,
> - correcting pseudocode wording and formatting issues.
>
> ### Closing
>
> We appreciate the reviewer’s feedback. The revised PDF incorporates many of the requested changes and, we hope, makes the paper substantially clearer and easier to evaluate.

---

### Review · Reviewer_pRW1 · 2026-04-03

**Summary Of Contributions:**

The paper extends linear combinatorial bandits to a multiplayer setting with information asymmetry, where each player independently controls an arm and coordination is only allowed before interaction begins. It introduces and analyzes three asymmetry regimes: action, reward, and combined; and shows that implicit coordination without communication is possible via mechanisms such as pre-agreed exploration orders or observable deviations. The authors derive near-optimal, gap-independent regret bounds of $\tilde{\cal{O}}(\sqrt{T})$ for all settings, significantly improving over prior gap-dependent results and matching single-player performance up to problem-dependent factors. Finally, they generalize the framework to multi-arm choices per player, showing only a modest $K^2$ overhead, highlighting that coordination rather then arm complexity, is the main challenge.

Strengths:
- The problem extensions are interesting and non-trivial
- The proposed solutions are elegant and achieve strong, gap-independent regret bounds
- The experimental results partially corroborate the theoretical findings, and the code is publicly available, supporting reproducibility
- The paper is generally well written

Weaknesses:
- Several definitions are either missing or introduced only after the corresponding terms are first used, which may confuse non-expert readers
- Most of the proofs are not included in the main body of the paper

**Additional Comments:**

None

**Audience:**

Yes

**Audience Explanation:**

As noted earlier, the problem extensions analyzed in the paper are both interesting and non-trivial (see other strengths mentioned above). Consequently, this research is likely to be of significant interest to the MAB community

**Claims And Evidence:**

Yes

**Claims Explanation:**

The correct answer is that it appears so, but I am not entirely certain, as most proofs are not included in the main body of the paper. The authors do provide the algorithms, which are generally clear and understandable, as well as intuition for some of their theorems and lemmas. However, to fully verify that all claims are supported, the appendix should also be reviewed

**Requested Changes:**

Requests:
- Enhance the intuitions and/or provide proof outlines for all theorems and lemmas in the main body of the paper
- Add a brief explanation of the "predefined order" when it is first mentioned
- Although “sabotage round” is used in prior work, consider replacing it with “signaling round” for clarity
- Reconsider the reward matrix example at the top of page 10, as it may not help the reader better understand the setting
- Address the typos listed below.

Typos:
- In the introduction, both $\cal{K}$ and $K$ are used to indicate multi-arms per player. Please unify the notation
- In Equation (2), the empirical reward mean $\hat{\mu}_a(t)$ is used before it is defined (definition appears later)
- Add spaces between the two parts of Equation (5)
- In the second part of Equation (5), it should be $\hat{\mu}_a(t)$ instead of $\hat{\mu}_a$
- In the last sentence of the paragraph following Equation (6), the words “The implementation is shown to the right” seem out of context
- Add a definition for the combinatorial arm gap $\Delta_{\vec{a}}$ when it is first mentioned
- The confidence interval at the bottom of page 6 should be dependent on $m$ and appears unused in the main body
- Remarks 1 and 2 are identical — consider merging or removing redundancy.

---

> ### Author Response · Authors · 2026-04-08
> **response to reviewer**
>
> # Response to Reviewer pRW1
>
> We thank the reviewer for their careful reading and constructive feedback. We have revised the manuscript to address the raised points; we summarize the changes below.
>
> **Clarity of definitions and notation.** We unified the notation for the number of arms, using $K$ consistently throughout (previously $\mathcal{K}$ appeared in the introduction). We added an explicit definition of the empirical mean $\hat{\mu}_{a}(t)$ before its first use in the UCB/LCB equations. The combinatorial arm gap $\Delta_{\vec{a}}$ is now defined in the Preliminaries rather than deferred to later sections. We also added a brief explanation of what the "predetermined ordering" entails when it is first introduced in the Coordination Mechanism paragraph.
>
> **Proof outlines in the main body.** We added proof sketches for the main theorems (Theorems 1, 4, and 5) to give readers intuition for the key arguments without requiring them to consult the appendix. For instance, the sketch for Theorem 1 outlines how the regret decomposes under the good event $G$, with suboptimal actions eliminated after $O(M^2 \log T / \Delta_{\vec{a}}^2)$ pulls, and the gap-independent bound following via Cauchy–Schwarz.
>
> **Terminology: "sabotage" → "signaling."** We replaced "sabotage round" with "signaling round" throughout the paper, which we agree better conveys the coordination mechanism.
>
> **Merging redundant remarks.** Remarks 1 and 2 have been consolidated into a single remark that covers both the role of the predetermined ordering and the importance of cyclic scheduling to prevent arm starvation.
>
> **Typos and formatting.** We corrected the notation issues (spacing in the good-event equation, ensuring $\widehat{\mu}_a(t)$ is used consistently). Regarding the sentence "The implementation is shown to the right" following Equation (6) in the appendix — we will verify and remove any stale references in the next revision pass.
>
> **Reward matrix example.** We will revisit the reward matrix example at the top of the joint-arm section to ensure it aids reader comprehension; we appreciate this suggestion.
>
> We believe these revisions substantially improve the paper's readability and thank the reviewer again for their detailed and helpful feedback.

---

### Decision · Action_Editor_GLJv · 2026-05-12

**Recommendation:** Reject

**Additional Comments:**

Reviewer reports indicate a general lack of rigor and verification of proofs, which may point to insufficient expertise of the authors. Abundant presentation issues further exacerbate this. Authors are encouraged to seek expert vetting of their analysis before resubmitting. Careful editing of the paper for clarity and accuracy are needed before this paper is ready for resubmission.

**Audience:**

Yes

**Audience Explanation:**

The paper analyzes a combinatorial multi armed bandit setting with information assymetry. This can be of interest to TMLR's audience.

**Claims And Evidence:**

No

**Claims Explanation:**

The reviewers have founds several mistakes in the proofs. While the authors have attempted to correct the mistakes, the general level of rigor is low and issues still exist.  In addition, presentation is lacking appropriate definitions, assumptions are not stated and notations are missing. Significant revision and verification of proofs are required before this paper can be considered for publication.

**Resubmission Of Major Revision:**

The authors may consider submitting a major revision at a later time.